



**Particulate matter (PM) episodes at a suburban site in Hong Kong: evolution of PM characteristics and role**

**of photochemistry in secondary aerosol formation**

Yi Ming Qin[1], Yong Jie Li[2,*], Hao Wang[3], Berto Paul Yok Long Lee[3], Dan Dan Huang[1], and Chak Keung Chan[1,3,4,*]

[1]Department of Chemical and Biomolecular Engineering, Hong Kong University of Science and Technology, Hong Kong, China

[2]Faculty of Science and Technology, University of Macau, Taipa, Macau, China

[3]Division of Environment, Hong Kong University of Science and Technology, Hong Kong, China

[4]School of Energy and Environment, City University of Hong Kong, Hong Kong, China

[*]To Whom Correspondence Should be Addressed

Chak K. Chan: AC1-G5716, School of Energy and Environment, City University of Hong Kong,

Tat Chee Avenue, Kowloon, Hong Kong, China

Tel: (852) 3442-5593; Fax: (852) 3442-0688

Email: chak.k.chan@cityu.edu.hk

Yong Jie Li: E11-3017, Faculty of Science and Technology, University of Macau, E11, Avenida da Universidade,

Taipa, Macau, China

Tel: (853) 8822-4943; Fax: (853) 8822-2426

Email: yongjieli@umac.mo





**Abstract**

2       Episodes with high concentrations of particulate matter (PM) across the seasons were investigated during four

one-month campaigns at a suburban site in Hong Kong. High-resolution time-of-flight aerosol mass spectrometer
(HR-ToF-AMS) measurements revealed that both regional transport and secondary formation contributed to high
PM levels during the episodes at this site. Based on distinct meteorological conditions, episodes were categorized
into three types: liquid water content (LWC), solar irradiance (IR), and long-range transport (LRT). Despite the
difference in meteorological conditions, all episodes were characterized by a high fraction of sulfate (45%-56%) and
organics (23%-34%). However, aerosols in LWC episodes were less aged, consisting of the lowest fraction of
secondary organics aerosols (SOA) and the highest fraction of small particles. Large particles mixed internally while
freshly formed small particles mixed externally in LWC episodes. Aerosols in LRT episodes, by contrast, were the
most aged and consisted of the highest proportion of low-volatility oxygenated organic aerosols (LVOOA) and the
lowest proportion of small particles. Both small and large particles mixed externally in LRT episodes. The highest
proportion of semi-volatile oxygenated organic aerosols (SVOOA) and a medium proportion of small particles were
observed in IR episodes. Both small and large particles were likely externally mixed during IR episodes. Unlike in
the other two types of episodes, in IR episodes aerosols experienced the most dramatic size increase and diurnal
variation, with a time lag between SVOOA and LVOOA and a gradual increase in carbon oxidation state ($\overline{OS}_c \approx$
$2 \times O{:}C{-}H{:}C$).  Five out of ten episodes were of the IR type, further reflecting the importance of this type of episode.
The evolution of aerosol components in one particular episode of the IR type, which exhibited a clear land-sea
breeze pattern, was examined in detail. Sulfate and SOA due to photochemical aging were very efficiently produced
during the course of six hours. The "fresh" SOA (SVOOA) was initially formed at a higher rate than the "aged"
SOA (LVOOA). The SVOOA transformed to LVOOA at the later stage of photochemical aging. This tranformation
was further supported by mass spectral analysis, which showed an increase in the most oxidized ion ($CO_2^+$) and
decreases in moderately oxidized ones ($C_2H_3O^+$, $C_3H_3O^+$ and $C_3H_5O^+$). By measuring the physical and chemical
properties of PM in a highly time-resolved manner, the current study was able to demonstrate the dynamic and
complex nature of PM transformation during high-PM episodes.





## 1. Introduction

Hong Kong and the rest of the Pearl River Delta (PRD) in China have been battling air pollution episodes as a
result of rapid economic development and urbanization in the region (Ho et al., 2003; Zhong et al., 2013).
Meteorological conditions may govern the regional and long-range transport of air pollutants to Hong Kong. For
example, northerly winds can bring pollutants from the inland areas to Hong Kong, and have been suggested to be
responsible for regional air pollution events in winter (Fang et al., 1999; Huang et al., 2014b, 2009). The majority of
earlier studies used filter sampling with a low time resolution of hours to days, and so were unable to track the
temporal chemical transformation in high particulate matter (PM) episodes. This limitation has hindered our
understanding of the dynamic nature of PM undergoing rapid chemical transformations. Such chemical
transformation can occur within short time periods (e.g., within a day), and so do other physicochemical properties
such as hygroscopic and optical properties. High-time-resolution chemical characterization techniques, for example
the Aerodyne high-resolution time-of-flight aerosol mass spectrometer (HR-ToF-AMS), offer a temporal resolution
of a few minutes. These techniques can thus provide valuable information on rapid changes in the PM composition,
facilitating more detailed analysis of pollution events (Decarlo et al., 2006). HR-ToF-AMS measurements also give
the size distributions of components (DeCarlo et al., 2008; Lee et al., 2013b). These data can reveal the origin,
formation and atmospheric processing mechanisms of PM (Seinfeld and Pandis, 2006; Shiraiwa et al., 2013), but
they remain under-utilized in most aerosol mass spectrometer (AMS) studies.
Secondary formation has been recognized as an important route leading to high PM concentrations worldwide
(Zhang et al., 2015a) and is the main culprit for haze episodes in cities across China (Huang et al., 2014a). Secondary
organic aerosol (SOA) has been shown to dominate over primary organic aerosol (POA) after a few hours of
photochemical aging, for instance, in Mexico City (Decarlo et al., 2010; Volkamer et al., 2006), Pasadena (Hayes et
al., 2013) and Tokyo (Takegawa et al., 2006). Semi-volatile oxygenated organic aerosol (SVOOA), which serves as a
proxy for fresh SOA, has been shown to transform to low-volatility oxygenated organic aerosol (LVOOA), which
serves as a proxy for aged SOA, in laboratory experiments (Alfarra et al., 2012; Jimenez et al., 2009). Such
transformation process may contribute substantially to the accumulation of PM, leading to episodic events that are
frequently observed in the fast-developing city clusters in China (Huang et al., 2012; Zhang et al., 2015b).
We conducted four one-month campaigns in each of the four seasons at the Hong Kong University of Science and
Technology (HKUST) Air Quality Research Supersite (AQRS) from May 2011 to February 2012 using an Aerodyne





HR-ToF-AMS for non-refractory $PM_1$ (PM with aerodynamic diameter less than 1 micron). We have previously
examined the general characteristics (Lee et al., 2013a; Li et al., 2013, 2015), seasonal trends (Li et al., 2015), aqueous-
phase process in the formation of oxygenated organics (Lee et al., 2013a; Li et al., 2013, 2015), organic sulfur
compounds (Huang et al., 2015), and particle hygroscopicity (Meng et al., 2014; Yeung et al., 2014). Most of these
studies did not explore in detail the evolution of PM characteristics on the timescale of hours. The current work thus
focuses on high PM episodes lasting a day or several days across the seasons. Specifically, we aim to investigate the
role of local photochemical activity in the formation and evolution of secondary aerosols based on the real-time
chemical composition and size distribution data.

**2.    Experimental Section**
**2.1.    Sampling Site and Measurements**
The sampling periods were from 25 April to 1 June 2011 (spring), from 1 September to 29 September 2011
(summer), from 28 October to 15 December 2011 (autumn), and from 19 January to 1 March 2012 (winter). HR-ToF-
AMS measurements were conducted at the HKUST AQRS (22°20'N, 114°16'E). The HR-ToF-AMS operating
procedure, data analysis and species determination have been discussed in Lee et al. (2013a), Li et al. (2015) and
Huang et al. (2015). Briefly, the AMS was operated alternatively between the V+PToF combined mode and the W-
mode for 5 min each. A collection efficiency of 0.5 was employed for measurements at this site, where the particles
have overwhelmingly dominant sulfate content (Aiken et al., 2009; Li et al., 2013). Concentrations of methanesulfonic
acid (MSA) and organosulfates (OS) were estimated by combining the V-mode data for total concentrations and the
W-mode data for high-resolution mass spectral analysis for specific ions (Huang et al., 2015). Mass spectra of organic
sulfur compounds were obtained from standards in laboratory experiments to support the W-mode data analysis
(Huang et al., 2015). We further calculated the particle liquid water content (LWC) by applying E-AIM II (Clegg et
al., 1998) to explore the effects of aqueous processing on $PM_1$ composition. Gaseous species ($CO$, $CO_2$, $SO_2$, $NO$,
$NO_2$, and $O_3$) were measured with standard gas analyzers (Teledyne API). Volatile organic compounds (VOCs) were
measured by gas chromatography (Synspec GC955). Meteorological parameters were measured by an automatic
weather station mounted on a tower right next to the supersite. Particle hygroscopicity and size distribution
measurements have previously been taken with a HR-ToF-AMS at this site (Cheung et al., 2015; Man et al., 2015;
Meng et al., 2014; Yeung et al., 2014), and direct reference to the resulting publications will be made where necessary.





### 2.2. Data analysis

#### 2.2.1. Criteria for an episodic event

The total non-refractory $PM_1$ (NR-$PM_1$) concentration showed little seasonal variation, with monthly averages ranging from 14.3 to 15.9 µg m$^{-3}$ as reported by Li et al. (2015). In this work, we defined episodic events according to the following criteria: 1) lasting for at least 24 hours; 2) daily NR-$PM_1$ average mass concentration exceeding 15 µg m$^{-3}$ (overall monthly averaged concentration); and 3) maximum concentration exceeding 30 µg m$^{-3}$. According to these criteria, 10 episodic events were identified in the campaigns as shown in Figure S1.

#### 2.2.2. Source apportionment

Following the results in Li et al. (2015), 72-hour backward air trajectory analysis and positive matrix factorization (PMF) analysis were performed. Briefly, the back trajectory analysis was run at an elevation of 300 m using the HYSPLIT-4.8 (Hybrid Single-Particle Lagrangian Integrated Trajectory) model developed by NOAA/ARL (U.S. National Oceanic and Air Administration/Air Resources Laboratory). We classified air masses affecting Hong Kong into long-, medium-, and short-range transport patterns for transport distances of 1000 km, between 500 and 1000 km, and less than 500 km, respectively (Su et al., 2015).

For PMF analysis, a four-factor solution with hydrocarbon-like organic aerosols (HOA), cooking organic aerosols (COA), SVOOA, and LVOOA was adopted as in Li et al. (2015). During episodic events, HOA and COA contributed insignificantly (less than 6% and 5% respectively) to total organic aerosol (OA). Since we are interested in SOA transformation, HOA and COA were combined into one POA factor in the following discussion.

#### 2.2.3. Size distribution

Size distributions of aerosol species are obtained with the HR-ToF-AMS when a set of pre-selected m/z is scanned as a function of the particle time of flight. We focus on the size distribution analysis of sulfate and organics because of their overwhelming dominance in NR-$PM_1$. The raw 10-min mass-size distributions of organics and sulfate during the episodic events were averaged over various time periods pertaining to the resolution (e.g. 24 hours, 1 hour). The average size distributions were then fitted by the peak fitting tool Multipeak Fit V2 provided by Igor Pro (Wavemetrics) using two log-normal peaks. The peaks were chosen such that the fit residuals were minimized. The two fitted peaks of the size distributions will hereafter be referred to as the small particle mode and the large particle mode. Fitting examples can be found in Figure S2. We will focus on the most important fitting parameters: the particle mass-mode





109 diameter (vacuum aerodynamic diameter, $D_{va}$) indicating the shift in particle size and the integrated peak areas of the

110 small particle mode and the large particle mode indicating the changes in mass concentrations of larger or smaller

111 particles.

112

113 **2.2.4. Photochemical age**

114 The photochemical age is useful for studying the extent of photochemical processing in an air mass. One way to

115 estimate the photochemical age ($\Delta t$) is by the ratio of a less reactive hydrocarbon to a more reactive one (Kleinman,

116 2003):

117
$$\Delta t = \frac{Ln(\frac{Cj}{Ci}*\frac{Ci0}{Cj0})}{(ki-kj)[OH]} \quad (1)$$

118 in which $C_i$ and $C_j$ are concentrations of hydrocarbons $i$ and $j$ at time $t$, whereas $C_{i0}$ and $C_{j0}$ are concentrations of

119 hydrocarbons $i$ and $j$ at time 0. The symbols $k_i$ and $k_j$ are their respective rates of reaction with hydroxyl radicals (OH).

120 Hofzumahaus et al. (2009) reported a high average OH concentration of $15 \times 10^6$ molecules cm$^{-3}$ around noon in the

121 PRD region, much higher than model predictions. Zhou et al. (2014) used an OH concentration of $5.2 \times 10^6$ molecules

122 cm$^{-3}$ in their calulation of photochamical age in the PRD region.  Lacking definitive estimates, we used a conservative

123 OH concentration of $1.5 \times 10^6$ molecule cm$^{-3}$ (Hayes et al., 2013; Mao et al., 2009) for the discussion of oxidation

124 trends in this study. The ratio of benzene to toluene has been widely used (El Haddad et al., 2013) because of their

125 similar emission sources and significantly different rates of reaction with OH radicals ($k_{benzene-OH}= 1.23 \times 10^{-12}$ cm$^3$

126 molecule$^{-1}$ s$^{-1}$; $k_{toluene-OH}= 6.0 \times 10^{-12}$ cm$^3$ molecule$^{-1}$ s$^{-1}$ at 298K) (Atkinson Roger, 2000). Because toluene reacts more

127 rapidly with OH radicals than does benzene, it is depleted more quickly, resulting in higher benzene/toluene ratios in

128 aged air masses. However, the photochemical age can deviate when fresh pollutants are added to an aged air mass.

129 Since fresh pollutants were insignificant after 10:00 (i.e. no significant peaks of benzenze and toluene after 10:00; see

130 Figure S3), we set the start time at 10:00 for the discussion of photochemical aging.

131 **3. Results and Discussion**

132 **3.1. Meteorological conditions and classification of episodes**




Table 1 summarizes the meteorological conditions, $PM_1$ concentrations, and the estimated LWC in the 10 high
PM episodes. All of the episodes involved air masses that originated over East Asia from the north, northeast or
northwest of Hong Kong, and swept over part of the PRD region before reaching the site. Seven of the 10 episodes
(E1-E7) were characterized by medium-range trajectories and the other three (E8-E10) by long-range trajectories.
The individual trajectories are shown in Figure S3. E1 and E2 had much lower solar irradiance (IR) but higher LWC
than the other episodes, which distinguished them from the other medium-range transport episodes. Thus, E1 and E2
were categorized as episodes of the LWC type (medium-range transport with high LWC and low IR) and the other
medium-range transport episodes as episodes of the IR type (medium-range transport with high IR and low LWC).
Li et al. (2013) referred to E1 and E2 as foggy episodes, while Li et al. (2013) and Lee et al. (2013a) referred to E3
as a hazy episode. The long-range transport episodes might be less associated with the local site-specific conditions
and were categorized as episodes of the LRT type.
High concentrations of PM can have a number of causes, including enhanced primary emissions (Ji et al., 2014),
concentrating effects due to a decrease in the height of the planetary boundary layer (Petäjä et al., 2016), regional
transport (Huang et al., 2009), as well as active secondary formation (Hayes et al., 2013). Local primary emissions
were not very significant at this site, as can be seen from the low contribution of POA (less than 6%) throughout the
whole campaign. Boundary layer dynamics on the high PM days can be a factor affecting PM concentration, but the
effects were likely minimal as the highest concentration was usually observed during the day at higher mixing heights
(Figure S4). Therefore, regional transport and active secondary formation would be the most probable causes for the
episodic events of high PM concentrations at this suburban site. More detailed meteorological conditions with
chemical characteristics in each episode can be found in Figure S4.
**3.2. Chemical characteristics of high PM episodes**
Figure 1 shows the chemical constituents of NR-$PM_1$ in the three types of episodes. It is apparent that sulfate
dominated in all types of episodes. In Hong Kong, sulfate is largely regarded as a major regional pollutant with little
spatial variability, as in the rest of the PRD (Hagler et al., 2006; LOUIE et al., 2005). Nitrate contributed less than 4%
in LWC episodes and IR episodes, but more than 7% in LRT episodes.  As LRT episodes occurred in wintertime, the
higher nitrate concentration was likely driven by gas-particle partitioning of ammonium nitrate to the particle at low
temperatures (Seinfeld and Pandis, 2006). Using the PMF-resolved SVOOA and LVOOA as proxies for fresh and



aged SOA respectively (Zhang et al., 2011), more details of OA can be revealed. SVOOA had higher contributions in
IR episodes, while LVOOA contributed roughly twice as much as SVOOA did in LRT episodes, because the air mass
was already quite aged when reached the site. LVOOA and SVOOA made similar contributions in LWC episodes.

163        Figure 2 shows the diurnal variations of the NR-PM$_1$ species, PMF-resolved organic factors, as well as O$_x$ (O$_3$+NO$_2$)

in these three types of episodes. SVOOA and LVOOA as well as O$_x$ increased during the day in IR episodes, with a
time lag between SVOOA and LVOOA. A similar time lag was also observed between SVOOA and LVOOA in the
Yangtze River Delta (YRD), another fast developing region of China (Huang et al., 2012). These delays may be the
result of conversions from fresh SVOOA to aged LVOOA in the afternoon. We explore such a possibility in Section
3.6. SVOOA and LVOOA both exhibit flat diurnal patterns in LWC episodes and LRT episodes.

169        Elemental analysis of OA (ratios of H:C, O:C, N:C, S:C and OM:OC) from the high resolution mass spectra

provides useful information to assess OA evolution. Recently, Canagaratna et al. (2015) used an updated (Improved-
Ambient) method to estimate O:C and H:C ratios, and reported 27% higher O:C ratios and 11% higher H:C ratios than
those estimated using the original (Aiken-Ambient) method. Recalculating the elemental ratios for the September
dataset using the updated method shows little difference from those obtained by simply applying the respective factors
of 1.27 and 1.11 to the O:C and H:C ratios (Figure S5). Hence, the O:C and H:C ratios in this study were corrected by
factors of 1.27 and 1.11, respectively, with Aiken-Ambient values reported in our previous studies. In the Van
Krevelen diagram (Heald et al., 2010; Ng et al., 2011) shown in Figure 3a, data points for LWC episodes (blue) fall
into a lower O:C region than do the data points for IR (red) and LRT episodes (green). Even though data points for IR
episodes and LRT episodes have similar slopes and intercepts in the Van Krevelen diagram, data points for IR episodes
had a much wider spread. These trends are also reflected in the diurnal patterns of carbon oxidation state ($\overline{OS}_c \approx 2 \times$O:C-
H:C) (Kroll et al., 2011) in Figure 3b. The $\overline{OS}_c$ diurnal pattern in LRT episodes was relatively flat, suggesting that
oxidized organics were mostly transported to the site with minor in-situ oxidation. The $\overline{OS}_c$ in IR episodes gradually
increased from 09:00 until 15:00. Similar trends were observed for O$_x$, LVOOA and to a less extent, SVOOA. With
all these combined, we believe that the local photochemical processing of OA was more likely at play in IR episodes
than the long-range transport of processed aerosols.

**3.3. Size distributions of sulfate and organics**



Figure 4 shows the peak fitting results of the type-averaged size distributions of organics and sulfate mass. The
mass-mode diameters ($D_{va}$) for both the small and large modes of organics and sulfate did not differ considerably
across the episode types (smaller than 5% differences). Within each type of episode, sulfate had a smaller portion of
small particle mode than organics did, indicating that sulfate was mostly aged while local activities resulted in some
fresh emissions of organics. LWC episodes had the largest small mode contribution and LRT episodes had the smallest,
providing additional support that particles were more aged and large particles dominated in LRT episodes while the
proportion of fresher small particles was higher in LWC and IR episodes due to stronger local influences.
Various studies have analyzed the particle mixing state based on single-particle instruments such as the aerosol
time-of-flight mass spectrometer (Healy et al., 2013, 2014; Yang et al., 2012) and the single-particle aerosol mass
spectrometer (Wang et al., 2015). Particle mixing state can also be inferred from particle size information obtained
with the AMS. The large mode diameters of organics and sulfate (Pearson's R value equals to 0.7) were strongly
correlated with a slope close to unity in LWC episodes (Figure 5), suggesting that organics and sulfate are likely
internally mixed in large particles. This internal mixing may occur during the process of local aqueous oxidation,
which has been thoroughly discussed in Li et al. (2013). In IR episodes, during which local photochemical oxidation
may have a more obvious influence, larger particles do not mix well internally (poor correlation between the large
mode diameters of organics and sulfate with $R_{pr} = 0.2$). As discussed in Section 3.2, in IR episodes, organics showed
a clear noontime peak associated with local photochemical activities while sulfate was still mainly a regional pollutant.
As a result, large particles of organics and sulfates were very likely externally mixed during IR episodes. Good
correlation ($R_{pr} = 0.7$) with a large deviation of the slope from unity (slope = 0.5) was observed in LRT episodes. As
long-range transport is the dominant process causing high PM levels during LRT episodes, organics and sulfate would
be brought to the site together, so their large mode diameters tend to be strongly correlated. However, they may have
different origins and may also have undergone different aging processes during the course of long-range transport,
and thus they would have different mode diameters. The correlations between the small mode diameters of organics
and sulfate were notably weaker, with $R_{pr}=0.5$ in LWC episodes, $R_{pr}=0.2$ in IR episodes, and $R_{pr}=0.2$ in LRT episodes,
suggesting that freshly formed small particles mixed externally.





Figure 6 shows that, in the two LWC episodes, the size variations for both the small and large modes were less
obvious: -2.5% for organics small mode, +8.1% for organics large mode, +1.6% for sulfate small mode, and -3% for
sulfate large mode from start to end as shown in the figure. By contrast, the size variations in the IR episodes were
much more drastic: +51.3% for organics small mode, +40.5% for organics large mode, +45.4% for sulfate small mode,
and +35.9% for sulfate large mode. Furthermore, particle sizes usually increased more rapidly during the days before
the IR episodes (days shaded in blue in Figure 6) than during the actual episodes (days shaded in orange). Since the
particles were fresher and smaller during the days before episodes, they grew more rapidly as gas phase semi-volatile
components condensed onto the particles. Moreover, the number concentrations of the pre-existing particles (acting
as condensation nuclei) were lower before than during the episodes, rendering more rapid size increases before the
episodes.

### 224    3.4. Frequency of high PM1 episodes

As is apparent from Table 1, the occurrence of different types of episodes exhibits a seasonal trend. LWC episodes
occurred only in spring and LRT episodes only in winter, while IR episodes took place in spring, summer and autumn.
This result is consistent with previous results (Huang et al., 2009) in that the frequency of high PM days in Hong Kong
had a strong seasonal variation. In winter, the overwhelming northerly wind brings pollutants via long-range transport
(Fang et al., 1999). In spring, foggy days with high PM levels are common due to the moisture-laden air masses
coming in from the sea and aqueous-phase processing of particulate species (Li et al., 2013). In summer, however,
hazy days are mainly due to high photochemical activities in this subtropical area, resulting in the formation of
secondary aerosols (Hu et al., 2008; Zhou et al., 2014).

### 233    3.5. Local photochemical formation and evolution of PM: A case study

### 234    3.5.1.    Time series of species during the local photochemical episode

Because of the high frequency of occurrence of IR episodes, we chose one typical IR episode (E4) to examine the
evolution of the aerosols with photochemical oxidation. This particular episode (E4) was under the influence of a clear
land-sea breeze pattern with weak winds (Figure 7), a typical meteorological phenomenon that affects air pollution
dynamics at this coastal city (Lee et al., 2013a). Despite the fact that the 72-hour air mass back trajectories (with
arriving height of 300 m) reflected medium-range transport, the surface air was quite stagnant as shown by wind speed





data collected from the ground station. As can be seen from Figure 7, the maximum wind speed was less than 2 m s$^{-1}$
while the average wind speed was approximately 0.5 m s$^{-1}$. The wind direction changed from northerly to easterly
between 06:00 and 10:00 and remained easterly until 20:00, when it changed clockwise from easterly back to northerly.
Under such conditions, local photochemical activities can lead to effective production and accumulation of air
pollutants. Time series of organics, sulfate, ammonium, nitrate, MSA, OS, PMF-resolved organic factors, some
gaseous species, as well as meteorological parameters were analyzed. Most NR-PM$_1$ species showed clear diurnal
variations. Figure 7 shows that organics increased from a roughly constant concentration of 10 µg m$^{-3}$ at night until
09:00 to its highest concentration of 16.6 µg m$^{-3}$ at 13:00, while sulfate showed a mild increase at 06:00 and then a
sharp increase at 10:30 to reach its highest concentration of 17.4 µg m$^{-3}$ at 16:00. They were overall consistent with
the increasing trend of irradiance, an indicator of photochemical activities, in the afternoon. Nitrate concentration was
high (2.5 µg m$^{-3}$) in the morning and started to decrease from 12:30 onwards to reach 0.3 µg m$^{-3}$ by 16:00, likely
attributable to vertical dilution due to a rise in the height of the planetary boundary layer, or alternatively evaporation
of ammonium nitrate at higher temperatures and lower RH values (Seinfeld and Pandis, 2006). Wind direction started
to change at 20:00, when all the NR-PM$_1$ species were at their lowest concentrations. POA concentration increased
from 2.5 µg m$^{-3}$ at 00:00 to about 5 µg m$^{-3}$ at 06:00, which might be due to the lowering of the planetary boundary
layer. Conversely, expansion of the boundary layer early in the morning could help disperse the POA. The increase in
LVOOA lagged behind that in SVOOA. Starting from 06:00, SVOOA concentration increased rapidly and peaked at
approximately 13:00, coinciding with the IR peak, possibly due to fresh SOA formation. LVOOA gradually increased
from 12:00 and peaked at 14:00, similar to sulfate. The time lag suggests that some conversion from fresh to aged
SOA might have occurred in the afternoon. Evaporation at the elevated temperature of 30°C throughout the afternoon
might also have led to the decrease in SVOOA, as with nitrate. The diurnal variation of MSA shows a noontime peak,
consistent with the trend of irradiance. In contrast, OS did not show a clear noontime peak, since OS at this site were
likely affected by inland transportation (Huang et al., 2015).
**3.5.2.   Changes in size distribution**
As shown Figure 8, before 06:00, the size distributions of sulfate and organics were both dominated by a mass
mode diameter of 500 to 600 nm. During 06:00-09:00, a shoulder at 200 nm appeared in the size distribution of sulfate
and in that of organics, indicating some fresh sulfate and organics were formed or emitted (possibly POA). As
photochemical reactions proceeded (09:00-18:00), the shoulder of D$_{va}$ at 200 nm became weaker and the size



distributions shifted to the larger end. It should be noted that during the whole aging process, the size distributions of
organics were broader than those of sulfate since organics were a mixture of numerous constituents from different
primary sources and reaction products formed via different atmospheric processes. The shifts in size distribution
suggest that secondary aerosol particles with sulfate and organics aged gradually and grew into larger particles.

### 3.5.3.    Photochemical production of secondary species

273        We examine the daytime photochemical activity during E4 by looking at the $SO_2$ oxidation and changes in the

degree of oxygenation of particulate organics. The sulfur oxidation ratio (SOR) has been used to evaluate the extent
of atmospheric oxidation of $SO_2$ to sulfate (Squizzato et al., 2013; Wang et al., 2005). Figure 9c shows the increase in
SOR from 0.2 at 9:00 to 0.7 at 18:00, indicating an efficient conversion from $SO_2$ to sulfate during daytime in this
episode. Figure 9b shows that the $\overline{OS}_c$ increased sharply near 11:00. $\overline{OS}_c$ was high after 18:00 because most of the
organics in PM had been converted to highly oxidized organic compounds during the aging process. Indeed, during
this period, LVOOA was the dominant OA component (Figure 7). The increases in SOR and $\overline{OS}_c$ coincided with the
increase in the ratio of benzene to toluene (Figure 9c). The oxidation of sulfur species and organic species reflects
efficient oxidation during this photochemical episode.

282        To semi-quantitatively evaluate the efficiency of SOA and sulfate formation, the changes in SOA/ΔCO,

MSA/ΔCO, and sulfate/ΔCO are plotted in Figure 10 as a function of photochemical age from 10:00 to 18:00. ΔCO,
defined as the measured CO concentration minus the minimum CO concentration (see Figure 7 for the time series of
CO), is assumed to be a conservative tracer of urban combustion emissions. The perturbations of CO concentration
by photochemical formation from VOC or destruction by OH radicals were thought to be negligible over such a short
timescale (less than eight hours) (Griffin et al., 2007). Normalization of species concentrations to the ΔCO
concentration is expected to reduce the effect of dilution (Hayes et al., 2013; Zhou et al., 2014).

289        From 10:00 to 18:00, sulfate/ΔCO increased by a factor of 7-8 as photochemical activity increased on a timescale

of approximately 6 h, with a formation rate (indicated by the slope of species/ΔCO vs. photochemical age) of
approximlately 48 μg m$^{-3}$ ppm$^{-1}$ h$^{-1}$. MSA/ΔCO also increased by a factor of approximately 3 at a rate of 0.05 μg m$^{-3}$
ppm$^{-1}$ h$^{-1}$ during photochemical aging. The good correlation of MSA production with the photochemical age suggests
that MSA originated from the reaction of gaseous dimethyl sulfide with OH radicals (Barnes et al., 2006). For
comparison, Bardouki et al. (2003) also found that MSA and OH radicals covaried over the northeastern coast of Crete.
As shown in Figure 10c, SOA/ΔCO increased by approximately a factor of 2 with the slope of 7.2 μg m$^{-3}$ ppm$^{-1}$ h$^{-1}$





(8.07 µg sm$^{-3}$ ppm$^{-1}$ h$^{-1}$). A shallower slope (approximately 4.0 to 4.5 µg sm$^{-3}$ ppm$^{-1}$ h$^{-1}$) was observed in Pasadena,
California from May to June (Hayes et al., 2013) while a similar slope (6.18 µg m$^{-3}$ ppm$^{-1}$ h$^{-1}$) was observed in a
previous study in Hong Kong in August (Zhou et al., 2014). This indicates that the SOA production in Hong Kong
during the local in situ photochemical oxidation in summer is high.
More interestingly, SVOOA/ΔCO increased during the first three hours but decreased slightly after 13:00, even as
photochemical age increased. In contrast, LVOOA/ΔCO increased steadily throughout the whole stage. After
photochemical processing for 6 h, LVOOA/ΔCO increased by approximately a factor of 20, from 2.3 µg m$^{-3}$ ppm$^{-1}$
to 49.4 µg m$^{-3}$ ppm$^{-1}$. Even though both SVOOA/ΔCO and LVOOA/ΔCO increased in the first stage, they did so at
slightly different rates, where SVOOA/ΔCO increased faster than LVOOA/ΔCO. This suggests that the production of
SVOOA was more efficient than that of LVOOA in the first stage. However, in the later stage of SOA formation, the
net productions of SVOOA were negative, which indicates that SVOOA may have photchemically converted to
LVOOA. As discussed earlier, the input of POA and VOC was limited to the early morning in our study. SVOOA
was consumed more quickly to form LVOOA than was replenished through further production in the late afternoon.
The situation where limited precursors exist to replenish fresh SOA (even under strong photochemical activity) might
also occur in other non-urban atmospheric environments, and thus may have an implication for OA transformation in
general.

### 3.5.4.    Mass spectral evolution

Figure 11a shows the evolving organic mass spectra during E4 (corresonding to the period of photochemical aging).
Eight spectra at one-hour intervals from 10:00 to 18:00 are shown from top to bottom. Two changes in the mass spectra
with photochemical processes were apparent: 1) decreases in the signal intensities of relatively high m/z ions (e.g.,
m/z 55, 57,67, 69, etc.), which indicates greater fragmentation (C-C bond cleavage) with photochemical oxidation;
and 2) increases in the mass concentrations of ions having m/z values of 28 (mainly $CO^+$) and 44 (mainly $CO_2^+$),
which presumably come from aldehyde, ketone and carboxylic acid (Ng et al., 2011). These changes are also reflected
in the relative intensity changes of hydrocarbon-like and oxygen-containing ions such as $C_4H_7^+$ , $C_2H_3O^+$ and $CO_2^+$
(Figure 11b, c). The  fractions of tracers of primary organic aerosols $C_3H_7^+$(m/z 43), $C_4H_7^+$ (m/z 55) and $C_4H_9^+$ (m/z
57) (Lambe et al., 2012) decreased. On the other hand, ion fractions of $C_2H_3O^+$ (m/z 43), $C_3H_3O^+$ (m/z 55) and $C_3H_5O^+$
(m/z 57) increased until 13:00 (corresponding to the peak of SVOOA), followed by the decrease of these moderately
oxygenated ions. These ions are predominantly from non-acid oxygenates, and are usually associated with fresh SOA.



However, The most oxidized ions, $CO_2^+$ (m/z 44), which is thought to be the marker of aged SOA, increased
continuously. As a result, the mass spectra, which were initially SVOOA-like, evolved to become LVOOA-like with
increasing photochemical age (Figure 11d). Overall, this spectral analysis indicates increasingly oxidized organics, as
long carbon chains became more functionalized and fragmented after extensive oxidation (Alfarra et al., 2012; Kroll
et al., 2009). Such an observation implies efficient transformation of OA within a few hours of photochemical aging,
a timescale that could be relevant to chemical transport models concerning SOA formation.
**4. Conclusion**
High-resolution HR-ToF-AMS measurements were taken during four one-month campaigns in suburban Hong
Kong to illustrate the evolution of high PM episodic events across the seasons. Three types of episodes, medium-
range transport with high particle liquid water content (LWC episodes), medium-range transport with high solar
irradiance (IR episodes), and long-range transport (LRT episodes), were captured based on synoptic meteorological
conditions. Which type of episode occurred depended on the season, with LWC episodes occurring only in spring
and LRT episodes only in winter, while IR episodes took place throughout the year except in winter. Sulfate was the
major constituent of NR-PM$_1$ during all episodic events. The contribution of secondary organic species, including
SVOOA and LVOOA, varied across episode types, with more SVOOA in the IR episodes and more LVOOA in the
LRT episodes. Unlike in the other two types of episodes, in IR episodes organics experienced the most dramatic
diurnal variation, with a time lag between SVOOA and LVOOA. This variation was associated with Ox, indicating
the conversions from fresh to aged SOA under photochemical oxidation. Elemental analysis involving the Van
Krevelen diagram and carbon oxidation state ($\overline{OS}_c \approx 2 \times O:C - H:C$) further showed that organics in IR were gradually
oxidized.
Fitted mass-mode diameters for both the small and the large mode of organics remained roughly constant across
episode types, while sulfate had a constant small mode diameter in all three types of episodes but a slightly
increased large mode diameter in IR episodes. The fraction of small particles decreased from LWC episodes to IR
episodes then to LRT episodes, suggesting that aerosols from long-range transport were more aged and dominated
by large particles while episodes under a greater influence of local processes had a higher proportion of fresher
small particles. Large particles mixed internally only in LWC episodes, and were more likely to mix externally in IR
and LRT episodes. Freshly formed small particles mixed externally in all types of episodes. In IR episodes, aerosols
underwent an obvious size increase, while in LWC episodes, the size increase was much less drastic.





Because of the high frequency of IR episodes, we picked one particular IR episode featuring land-sea breeze to
examine in detail the evolution of aerosol components. Photochemical aging led to mode size shifting for sulfate and
organics, indicating particle growth. Increases in the sulfur oxidation ratio and carbon oxidation state were also
observed as the aerosols became more aged, which indicates that secondary inorganic species sulfate and SOA were
very efficiently produced within six hours of photochemical aging. In the earlier stage of aging, "fresh" SOA—
SVOOA—was formed at a higher rate than "aged" SOA—LVOOA. SVOOA clearly transformed to LVOOA at the
later stage of photochemical aging, resulting in a 20-fold increase in LVOOA. This conversion was further supported
by mass spectral analysis, which showed an increase in the most oxidized ion ($CO_2^+$) and decreases in moderately
oxidized ones ($C_2H_3O^+$, $C_3H_3O^+$ and $C_3H_5O^+$). With real-time size-resolved chemical composition data, we
demonstrated that aerosol components can transform very efficiently in just a few hours, a process that is essential in
understanding the dynamic nature of aerosol evolution during episodes with high PM concentrations.
**Acknowledgements**
This work was supported by the Environmental Conservation Fund of Hong Kong (project number ECWW09EG04)
and the Research Grants Council of the Hong Kong Special Administrative Region, China (General Research Fund
600413). Y.J. Li gratefully acknowledges the support from the Start-up Research Grant (SRG2015-00052-FST) of the
University of Macau. The grant from the HKUST Asian Future Leaders Scholarship Program (AFLSP) is also
gratefully acknowledged.

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




**Tables:**

*Table 1 Synopsis of meteorological conditions of high PM episodes*

| Episode | Season | Date | Air mass origin | Wind speed (m s⁻¹) | Solar Irradiance (w m⁻²) | Liquid water content (µg m⁻³) | PM₁ (µg m⁻³) | PM₁ Max (µg m⁻³) | Type |
|---|---|---|---|---|---|---|---|---|---|
| E1 | Spring | 28-30 Apr | M-R[a]/NE[b] | 0.7±0.4 | 41±67 | 47.1±15.9 | 25.5±3.1 | 33.1 | LWC |
| E2 | Spring | 14-16 May | M-R/NE[b] | 1.1±0.8 | 27±61 | 38.6±14.5 | 18.8±6.4 | 32.4 | LWC |
| E3 | Spring | 27-29 May | M-R[a]/NE[b] | 0.9±0.8 | 184±263 | 19.3±9.2 | 28.4±12.6 | 64.1 | IR |
| E4 | Summer | 2 Sep | M-R[a]/NW[b] | 0.5±0.4 | 111±163 | 20.0±3.1 | 22.5±6.1 | 33.7 | IR |
| E5 | Summer | 20-24 Sep | M-R[a]/NE[b] | 2.2±0.5 | 143±234 | 14.9±4.6 | 23.8±4.8 | 35.9 | IR |
| E6 | Autumn | 3 Nov | M-R[a]/NE[b] | 1.3±0.5 | 174±271 | 12.8±5.9 | 15.6±6.2 | 30.0 | IR |
| E7 | Autumn | 13-15 Nov | M-R[a]/NE[b] | 1.2±0.5 | 150±221 | 19.4±7.0 | 23.4±7.0 | 45.2 | IR |
| E8 | Winter | 24-25 Nov | L-R[a]/NE[b] | 1.6±0.5 | 112±174 | 14.1±6.6 | 25.9±6.2 | 38.6 | LRT |
| E9 | Winter | 8 Feb | L-R[a]/N[b] | 2.2±0.6 | 49±74 | 27.8±2.8 | 29.7±8.1 | 41.6 | LRT |
| E10 | Winter | 18-19 Feb | L-R[a]/NE[b] | 1.5±0.6 | 104±170 | 16.0±5.3 | 25.5±9.4 | 64.9 | LRT |

a: Range of air mass orgin: Medium range (M-R); Long range (L-R).
b: Direction of air mass origin: Northeast (NE); Northwest(NW); North (N).





**Figures:**

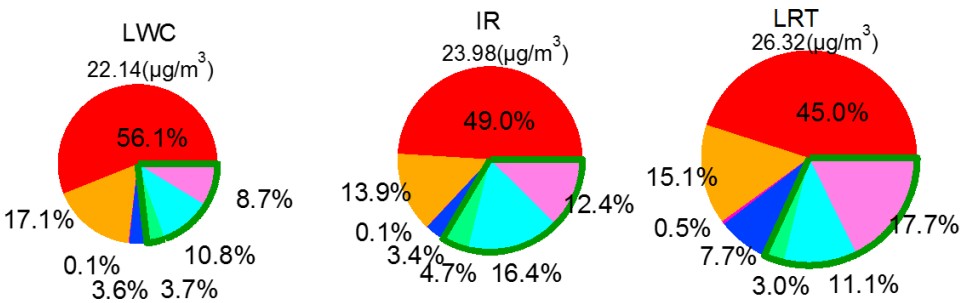

*Figure 1 Chemical constituents NR-PM$_1$ in LWC, IR and LRT episodes. (LWC: medium-range transport with high*
*LWC and low IR; IR: medium-range transport with high IR and low LWC; LRT: long-range transport)*





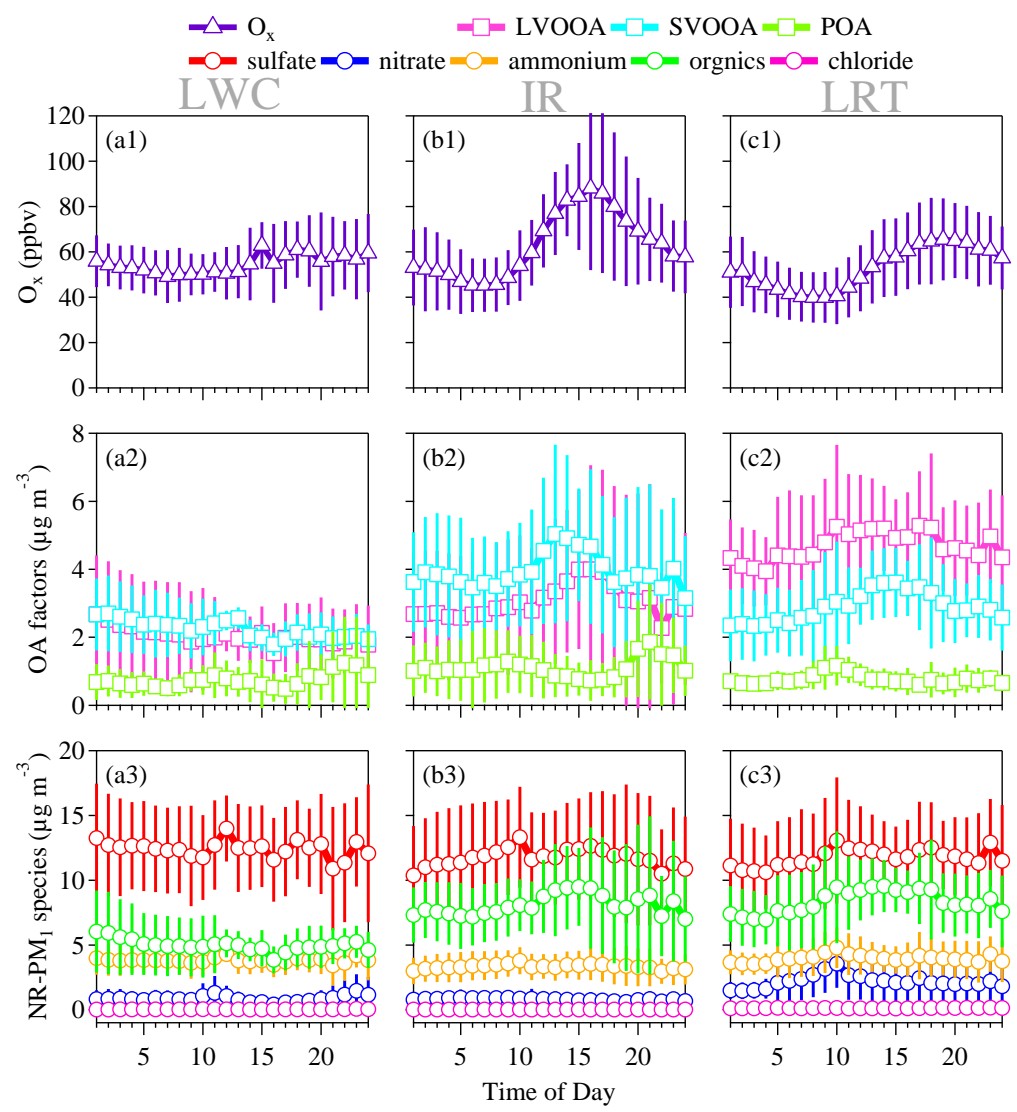

*Figure 2 Summary of diurnal variations of the PM₁ species, PMF-resolved organics as well as Oₓ in the three types*
*of episodes. Means are shown as points and standard deviations are as error bars.*





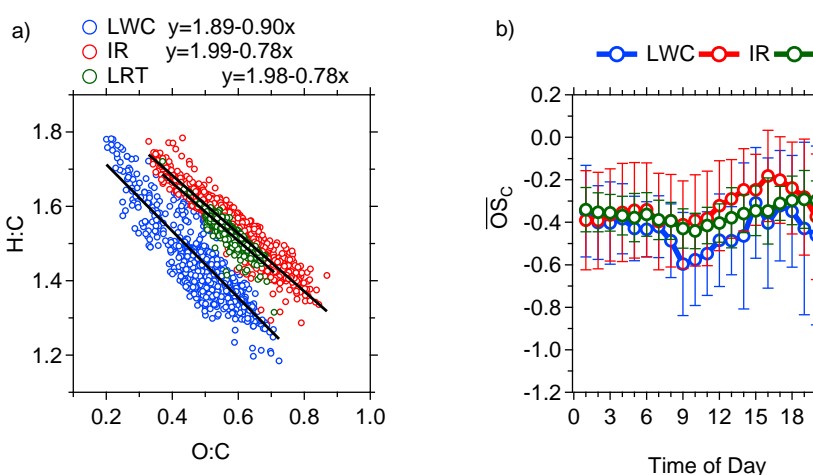

*Figure 3 a) Van Krevelen diagram for the three types of episodes; b) diurnal variation of carbon oxidation state*
*($\overline{OS_c}$ ). Means appear as circles with superimposed standard deviations.*





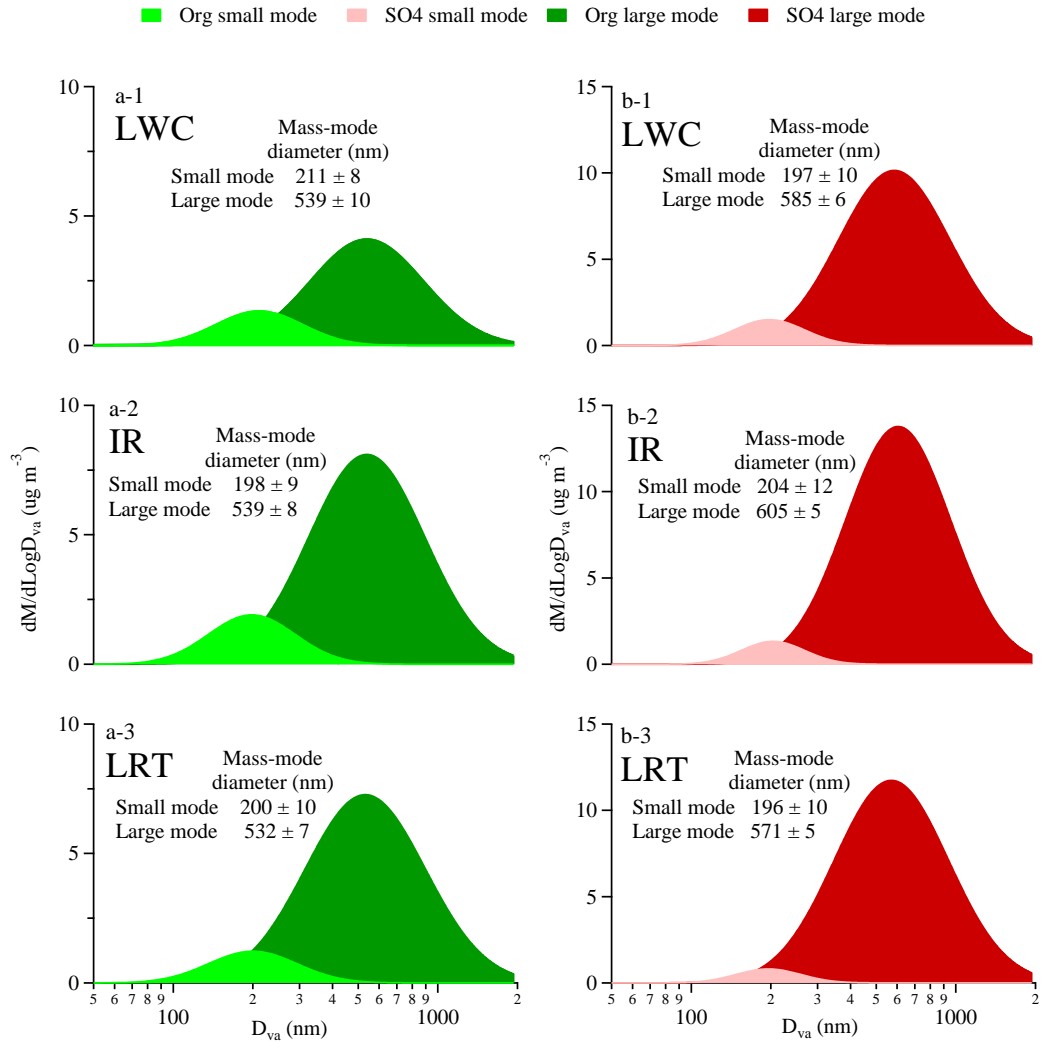

*Figure 4 Bimodal log-normal fitting results of the size distributions of organics and sulfate during the three types of episodes. a) Fitted small particle size mode and large particle size mode of organics during LWC, IR and LRT episodes; b) fitted small particle size mode and large particle size mode of sulfate during LWC, IR and LRT episodes.*



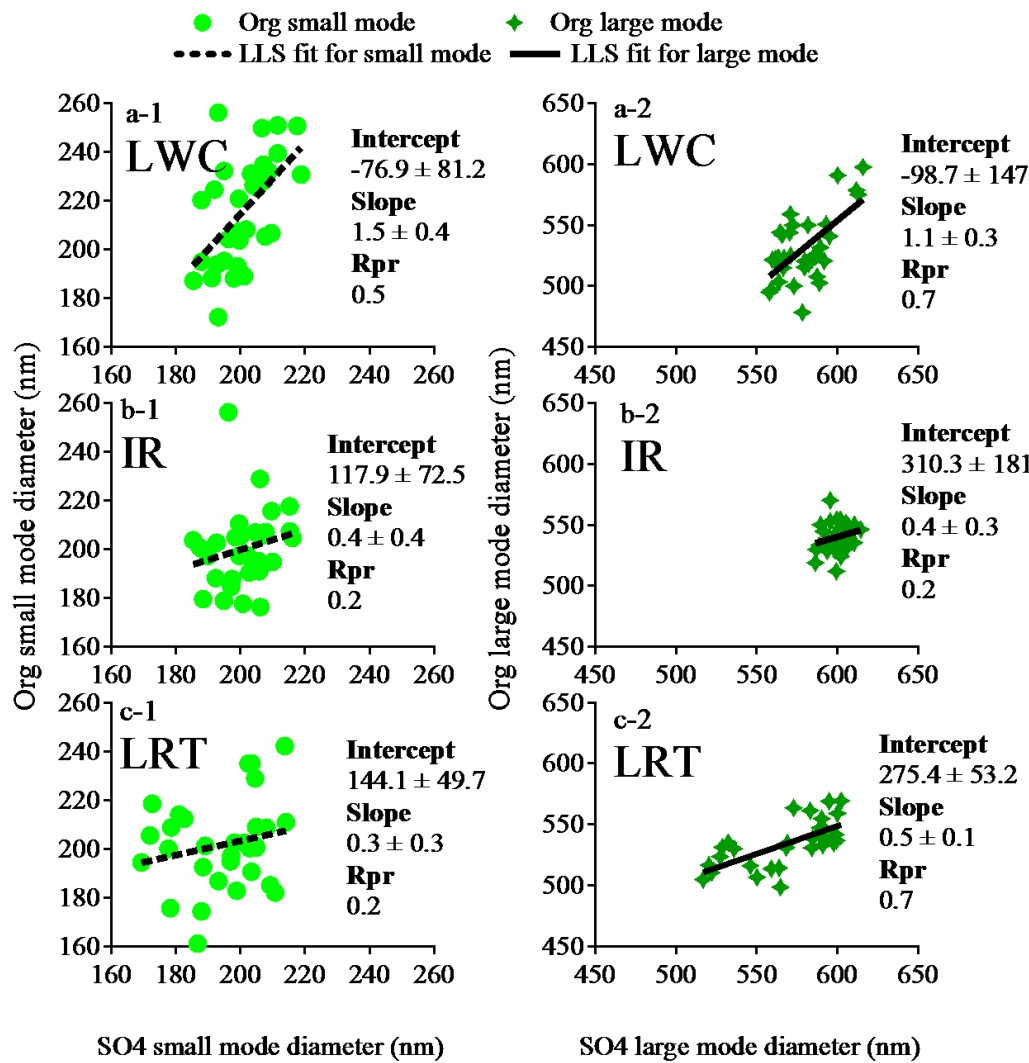

*Figure 5 Scatter plots and linear least square fits of mass-mode diameters of organics and sulfate during the three*
*different types of episodes. a1-a2) small and large mass-mode diameter of organics against sulfate during LWC*
*episodes; b1-b2) small and large mass-mode diameter of organics against sulfate during IR episodes; c1-c2) small*
*and large mass-mode diameter of organics against sulfate during LRT episodes.*



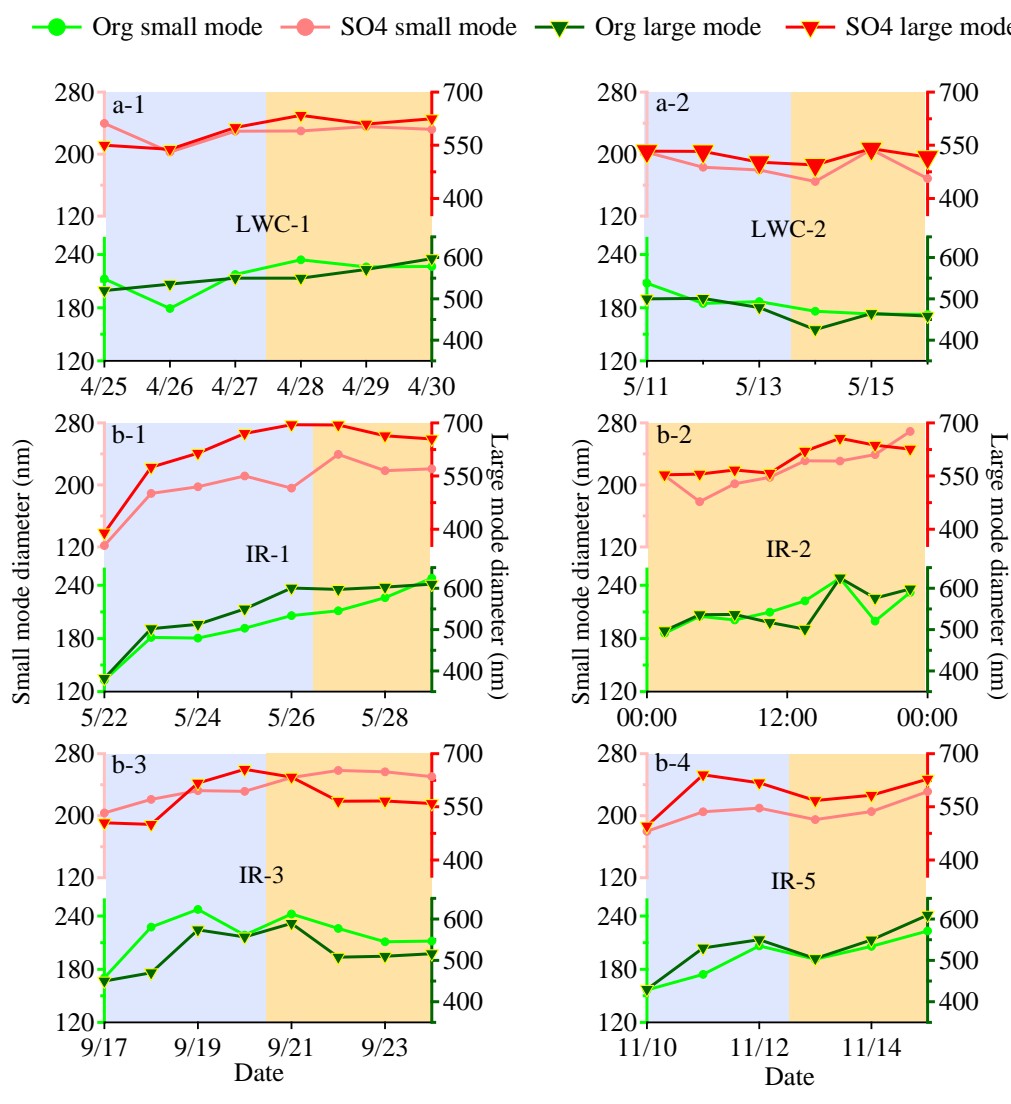

Figure 6 Variations of 24-hour averaged size distributions of fitted mass-mode diameters of organics and sulfates
during LWC episodes and IR episodes (shaded in orange) and several days before each episode (shaded in blue).
For the episode that lasted only for a day (E4), 3-hour averaged size distributions of fitted mass-mode diameters are
shown instead. a) LWC episodes; b) IR episodes.





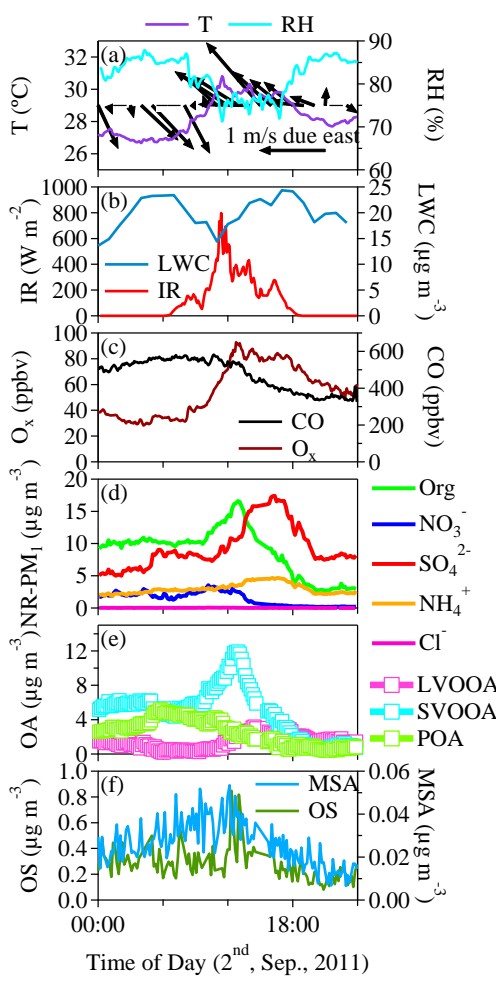

*Figure 7 Time series of meteorological parameters, gaseous species, NR-PM1 species and PMF-resolved organic*
*factors in E4.*







Figure 8 Size distributions of sulfate (A) and organics (B) in different time intervals during E4.



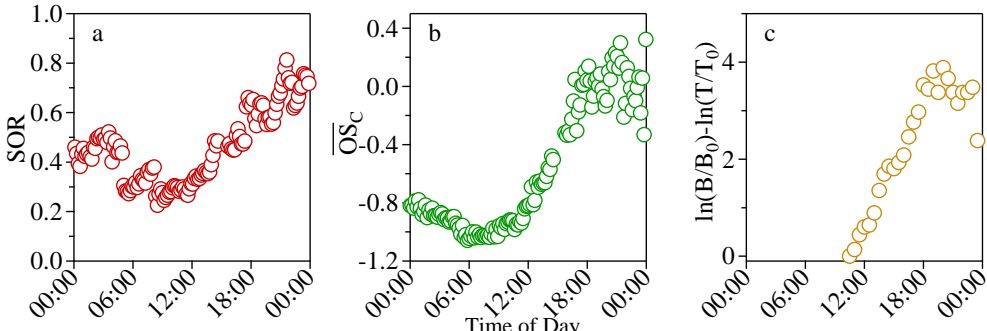

*Figure 9 Oxidative evolution of aerosol components. a) Sulfur oxidation ratio (SOR); b) average carbon oxidation state $\overline{OS_c}$; c) benzene to toluene ratio (B: benzene concentration at time t; B0: benzene concentration at time 0; T: toluene concentration at time t; T0: toluene concentration at time 0).*





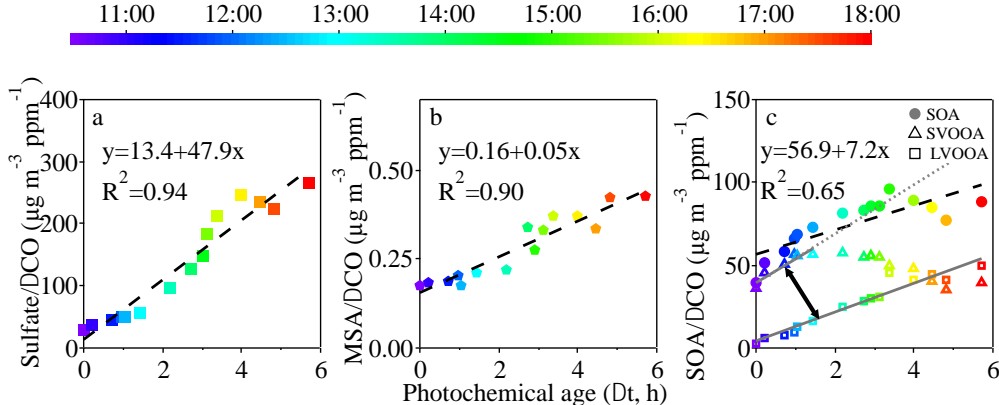


*Figure 10 Photochemical production of secondary species. a ) ΔCO-normalized sulfate concentration (SO4/ΔCO)*
*as a function of photochemical age; b) ΔCO-normalized MSA concentration (MSA/ΔCO) as a function of*
*photochemical age; and c) ΔCO-normalized secondary organic aerosol concentration (SOA/ΔCO,*
*SOA(SVOOA+LVOOA)) as a function of photochemical age. Data points are colored by time of day. Data points*
*represent half-hour averages.*



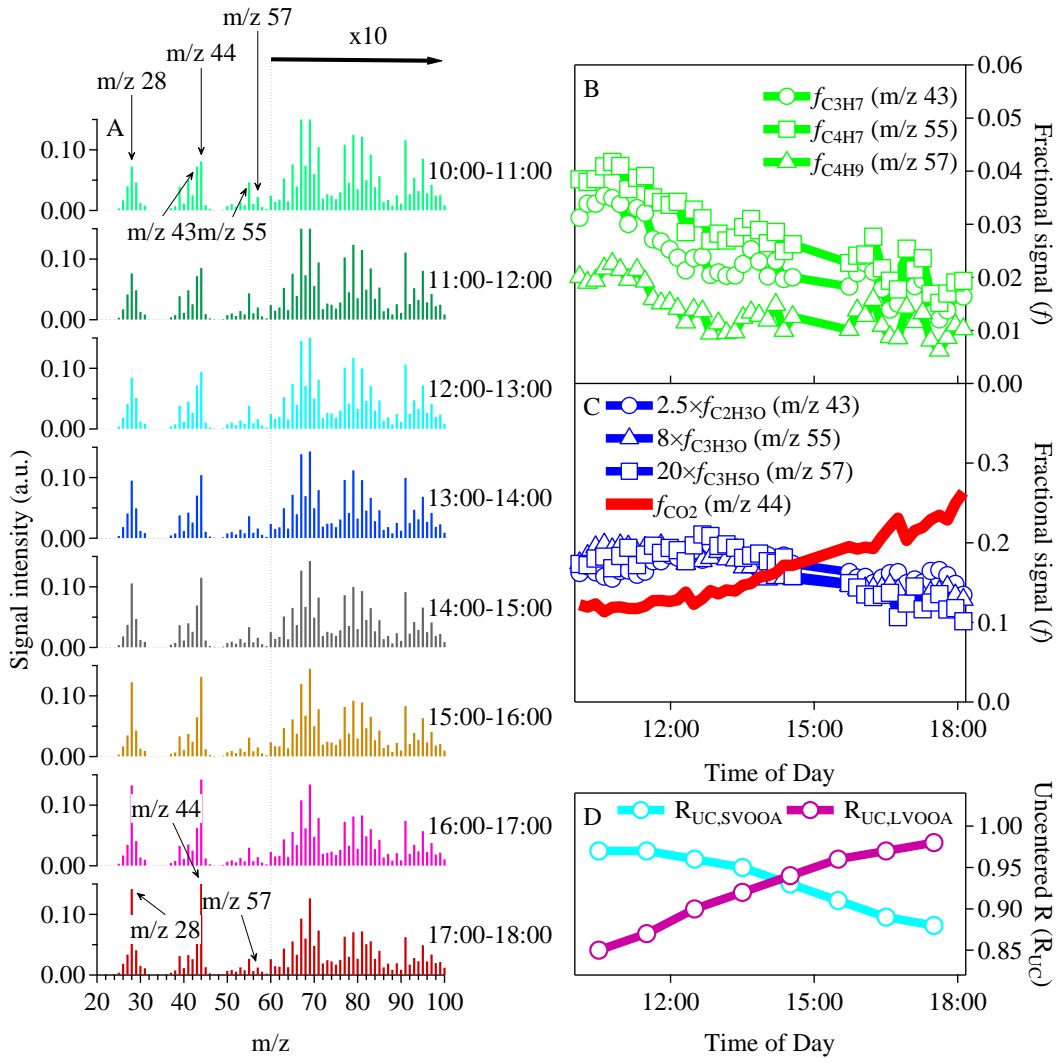


*Figure 11 Evolution of high-resolution organic mass spectra from 10:00 to18:00 during the photochemical aging process in E4: a) mass spectral evolution; b) changes in relative intensities of hydrocarbon-like ions $C_3H_7^+$ (m/z 43), $C_4H_7^+$ (m/z 55) and $C_4H_9^+$ (m/z 57); c) changes in relative intensities of oxygen-containing ions: $C_2H_3O^+$ (m/z 43), $C_3H_3O^+$ (m/z 55), $C_3H_5O^+$ (m/z 57) and $CO_2^+$ (m/z 44); and c) correlation of OA mass spectra with reference (Mohr et al., 2012) SVOOA and LVOOA mass spectra.*