# Peer review of "Particulate matter (PM) episodes at a suburban site in Hong Kong: evolution of PM characteristics and role of photochemistry in secondary aerosol formation"

_Atmospheric Chemistry and Physics, 2016_

## Referee Comment (RC1) · Anonymous Referee #1 · 28 Jul 2016

The manuscript of Y. M. Qin et al. presents the characteristics of fine PM in Hong Kong (chemical composition, size distribution, degrees of oxygenation, photochemical production and organic mass spectra) based on high resolution time-of-flight aerosol mass spectrometer (HR-ToF-AMS) measurements, positive matrix factorization (PMF) and air mass back-trajectory analyses. The manuscript is well structured and presented in a clear way. The methods used (PMF, back-trajectory analysis) are not innovative but are valid and well applied. The conclusions of this manuscript (e.g. the influence of meteorological conditions, particle mixing state, and the importance of photochemical aging on SOA formation in areas with high PM concentrations) are of scientific sig-

nificance. I therefore recommend publishing this work in Atmospheric Chemistry and Physics after the authors respond to the following comments.

Specific comments: 1. Page 3, Line 47-49: it should be more careful for the statement that SVOOA is a proxy for fresh SOA while LVOOA is aged SOA. There isn't an absolute relationship between the "fresh" / the "aged" (terms involved in the OA formation and evolution) and the volatility / oxidation state (terms for OA physicochemical properties). The description in Page 8744 of Li et al. (2013) could be considered to be adopted in this study: "less-oxidized OA (e.g. SVOOA)" and "more-oxidized OA (e.g. LVOOA)". Other statements about "fresh" SOA (SVOOA) and "aged" SOA (LVOOA) in other parts of the text (e.g. abstract) are suggested to be adjusted.

2. Page 9, Line 192-193: "aged" is an ambiguous term and the ageing process includes several specific mechanisms. I suggest that "aged" is replaced by more specific descriptions. In addition, as you stated that small particle proportions are higher in both LWC and IR episodes, are the underlying reasons/mechanisms the same for both LWC and IR episodes? More specific discussion is suggested to replace "stronger local influences".

3. Page 9, Line 194-197: The accuracy of the derived particle mixing state from AMS is suggested to be discussed with comparing with single-particle instruments. Line 211: comparison with previous studies about the mixing state in Hong Kong will be helpful.

4. Page 10, Line 213-217: How those percentage numbers are calculated for the size variations? Line 219-220: How do you derive this result that "Since the particles were fresher and smaller during the days before episodes, they grew more rapidly as gas phase semi-volatile components condensed on the particles"? Is condensation the only contributor to the particle growth? Line 220-221: Please show evidence to support "the number concentrations of the pre-existing particles were lower before than during the episodes". Why the size variations for the LRT episodes are overlooked? What is your object to show the sizes variations? What are the implications?

[Figure]

5. Page 10, Section 3.4 could be merged into Section 3.1 or just after the Section 3.1 (moving Section 3.2 to 3.3...).

6. Page 12, Line 275: Please replace Figure 9c by Figure 9a.
* * *

---

## Referee Comment (RC2) · Anonymous Referee #2 · 23 Aug 2016

General comments: The authors showed scientific findings on role of photochemistry in the evolution of secondary PM characteristics base on analyses of measurement data with high temporal resolution. The manuscript is generally well written and constructed, and therefore may be suitable for future publication in Atmospheric Chemistry and Physics after revisions based on referees' and other relevant comments. In addition, I recommend the authors to highlight the remarkable progress from their previous studies because they have already published several papers on the measurement data used in this study.

[Figure]

Specific comments: Page 6, Line 129-130: How can I find "no significant peaks of benzene and toluene after 10:00" from Fig. S3?

Page 7, Line 147: Is the quite low fraction of POA even in cool season common in and around the study region, or specific at the study site (period)?

Page p. Line 197-199: If organics and sulfate are internally mixed, diameters of organics and sulfate are strongly correlated. However, is the strong correlation sufficient evidence for the internal mixing?

Page 9, Line 199: Does the authors consider that fraction of sulfate was higher in LWC episodes (56%) than the other episodes because of efficient aqueous oxidation?

Page 10, Line 214-217: Please clarify how the percentages were estimated. In Fig. 5, the range of sulfate particle diameter seems to be smaller in IR episodes than LWC episodes.

Page 10, Line 235: Can discussion on the "typical" IR episode (E4) be generally applicable to the other four IR episodes in different season? Although temporal variation patterns in E4 may be similar to those in Fig. 2b, temporal variation patterns and ratio of LVOOA:SVOOA seems to be quite different from one IR episode to another (Fig. S4).

Page 10, Line 238-243: Does this mean that 72-h backward trajectory-based air mass origin could not explain the ground-level pollution, or the trajectories (at least for the first 24h) reflected a land-sea breeze pattern?

Page 11, Line 274-277: SOR was the maximum after 18:00 on the day. Is SOR appropriate indicator of the photochemical SO2 conversion?

Technical corrections: Page 6, Line 129: "benzenze" => "benzene"

Page 10, Line 224-232: "3.4. Frequency of high PM1 episodes" can be moved into "3.1. Meteorological conditions and classification of episodes".

Page 13, Line 313: "corresonding" => "corresponding"

Fig. S4: Why pressure decreases with increasing mixing height? It is hard to believe that pressure was frequently less than 900 hPa at the ground-level of the study site. Air temperature in 05/14-05/16 episode seems to be quite low. Is it correct? Please provide time series of gaseous species other than OX (i.e., CO, SO2 and NOX).

---

## Author Comment (AC1) · 26 Sep 2016

We thank the reviewers for the thorough comments and suggestions to improve the manuscript. Below we reply to the reviewer's comments point by point. We list the comments in black, our replies in blue, original text for change in red and revised text in green.

**Anonymous Referee #1**

**General comments**

The manuscript of Y. M. Qin et al. presents the characteristics of fine PM in Hong Kong (chemical composition, size distribution, degrees of oxygenation, photochemical production and organic mass spectra) based on high resolution time-of-flight aerosol mass spectrometer (HR-ToF-AMS) measurements, positive matrix factorization (PMF) and air mass back-trajectory analyses. The manuscript is well structured and presented in a clear way. The methods used (PMF, back-trajectory analysis) are not innovative but are valid and well applied. The conclusions of this manuscript (e.g. the influence of meteorological conditions, particle mixing state, and the importance of photochemical aging on SOA formation in areas with high PM concentrations) are of scientific significance. I therefore recommend publishing this work in Atmospheric Chemistry and Physics after the authors respond to the following comments.

**Specific comments**

1.  Page 3, Line 47-49: it should be more careful for the statement that SVOOA is a proxy for fresh SOA while LVOOA is aged SOA. There isn't an absolute relationship between the "fresh" / the "aged" (terms involved in the OA formation and evolution) and the volatility / oxidation state (terms for OA physicochemical properties). The description in Page 8744 of Li et al. (2013) could be considered to be adopted in this study: "less-oxidized OA (e.g. SVOOA)" and "more-oxidized OA (e.g. LVOOA)". Other statements about "fresh" SOA (SVOOA) and "aged" SOA (LVOOA) in other parts of the text (e.g. abstract) are suggested to be adjusted.

    Agree. Changes have been made as suggested. We adopt descriptions in Li et al. (2013).

2.  Page 9, Line 192-193: "aged" is an ambiguous term and the ageing process includes several specific mechanisms. I suggest that "aged" is replaced by more specific descriptions. In addition, as you stated that small particle proportions are higher in both LWC and IR episodes, are the underlying

reasons/mechanisms the same for both LWC and IR episodes? More specific discussion is suggested to replace "stronger local influences".

We agree that "aged" is an ambiguous term. We aim at distinguishing "aged" vs. fresh. However, it is hard to distinguish the underlying mechanisms of ageing based on the type-averaged size distributions of organics and sulfate mass only. Although the fractions of small particles are higher in both LWC and IR episodes than in LRT episodes, the underlying mechanisms may be different. We have discussed the possible mechanisms combining with the other observational results (e.g. diurnal variations, elemental analysis of OA, mixing state and size variations before and during these episodes) in the original manuscript. We have clarified it in the revised manuscript as below.

Change from:

Within each type of episode, sulfate had a smaller portion of small particle mode than organics did, indicating that sulfate was mostly aged while local activities resulted in some fresh emissions of organics. LWC episodes had the largest small mode contribution and LRT episodes had the smallest, providing additional support that particles were more aged and large particles dominated in LRT episodes while the proportion of fresher small particles was higher in LWC and IR episodes due to stronger local influences.

To:

Within each type of episode, sulfate had a smaller fraction of small particle mode than organics did, indicating that sulfate was relatively aged while organics received contributions from local fresh emissions. LWC episodes received the largest contribution from small mode sulfate because of some local influences whereas LRT episodes received the smallest contribution with relatively little local activities. The oxidation mechanisms, however, might be different. Aqueous phase oxidation may dominate in LWC episodes while photochemical oxidation may dominate in IR episodes.

3. Page 9, Line 194-197: The accuracy of the derived particle mixing state from AMS is suggested to be discussed with comparing with single-particle instruments. Line 211: comparison with previous studies about the mixing state in Hong Kong will be helpful.

Thanks for the insightful suggestion. It would be very useful to compare the data of the AMS with single-particle measurements. However, we did not have a single particle spectrometer at our disposal in this campaign. We admitted that this is one of the limitations of the current study.

In this study, we infer the mixing state of the components based on overlapping of their size distributions, similar to Bahreini et al.(2003). It is understood that overlapping of size distributions is a necessary but not sufficient condition for internal mixing. Hence, our discussion of the mixing states is qualitative but not quantitative. To our best knowledge, direct measurements of mixing state from single particle instruments across different seasons in Hong Kong have not been reported. Based on the AMS size distributions, we found that sulfate and organics in large particles were mixed internally only in LWC episodes, and were more likely to be externally mixed in IR and LRT episodes. Freshly formed small particles are externally mixed in all types of episodes. We hope that this study may add to a better understanding of organic and sulfate mixing states in Hong Kong. We have added the above discussions in the revised manuscript.

4. Page 10, Line 213-217: How those percentage numbers are calculated for the size variations? Line 219-220: How do you derive this result that "Since the particles were fresher and smaller during the days before episodes, they grew more rapidly as gas phase semi-volatile components condensed on the particles"? Is condensation the only contributor to the particle growth? Line 220-221: Please show evidence to support "the number concentrations of the pre-existing particles were lower before than during the episodes". Why the size variations for the LRT episodes are overlooked? What is your object to show the sizes variations? What are the implications?

The calculation of size variation and the number concentration have been clarified in the revised supplementary information. We did not focus on the size variations of the LRT episodes because it is difficult to assess what has happened to the particles before they reached the site based on measurements at the site. The particles were already aged when they reached the site. With little local aging, the size variation analysis may not give us further information on the characteristics of ageing. Instead, we would like to focus on LWC and IR episodes which have shown signs of local atmospheric processing.

Change from:

Figure 6 shows that, in the two LWC episodes, the size variations for both the small and large modes

were less obvious: -2.5% for organics small mode, +8.1% for organics large mode, +1.6% for sulfate small mode, and -3% for sulfate large mode from start to end as shown in the figure. By contrast, the size variations in the IR episodes were much more drastic: +51.3% for organics small mode, +40.5% for organics large mode, +45.4% for sulfate small mode, and +35.9% for sulfate large mode. Furthermore, particle sizes usually increased more rapidly during the days before the IR episodes (days shaded in blue in Figure 6) than during the actual episodes (days shaded in orange). Since the particles were fresher and smaller during the days before episodes, they grew more rapidly as gas phase semi-volatile components condensed onto the particles. Moreover, the number concentrations of the pre-existing particles (acting as condensation nuclei) were lower before than during the episodes, rendering more rapid size increases before the episodes.

To:

As discussed earlier, there may be some local atmospheric processing of aerosols in LWC and IR episodes but not in LRT episodes. Therefore, we further explored the mechanisms underlying the atmospheric processing of LWC and IR episodes based on the size variations before and during episodic events. Figure 6 shows the particle mass mode diameters and areas (concentrations) in the LWC and IR types of episodes. We obtained the percentage changes in mode diameters by comparing the smallest diameter before the episode and the largest diameter during the episode for each episode. These percentages in each episode was then averaged to obtain the percentage changes for each episode type. The results show that the changes in mode diameter were small in the LWC episodes: -2.5% for small mode organics, +8.1% for large mode organics, +1.6% for small mode sulfate, and -3% for large mode sulfate. In contrast, the changes in mode diameters changes were much more drastic in the IR episodes: +51.3% for small mode organics, +40.5% for large mode organics, +45.4% for small mode sulfate, and +35.9% for large mode sulfate. Furthermore, particle size usually increased more rapidly before the IR episodes (shaded in blue in Figure 6) than during the episodes (shaded in orange). With fewer pre-existing particles before the episodes, particle growth likely via condensation and reactive uptake of semi-volatile components was more rapid than during the episodes. The number concentration is discussed in detail in SI.

 SI:

Figure S7 shows the variations in the 24-h average of the fitted mode diameters and peak areas (and

indicator of mass concentration in that mode) of sulfate and organics during IR episodes. All mode diameters and the peak areas increased from before the episode (the blue shaded area) to during the episode (the orange shaded area). The extent of the peak area increase (mass concentration increase) exceeds what can be explained by the increase in mode diameter alone assuming spherical particles and constant number concentrations. As shown in Table S1, the increases in peak areas are always larger than the increase in the cube of mode diameters for both modes. This suggests an increase in the number concentration of particles during the episodes. The lower number concentrations of pre-existing particles before the episodes renders more rapid increases in size than during the episodes.

[Figure]

Figure S1' Variations of 24-h averaged of the fitted mode diameters and peak areas of organics (1)
and sulfates (2) before (shaded blue) and during (shaded orange) each IR episode.

Table S1. Mode diameters and peak areas before and during episodic days in each of the three IR episodes.

| | | Before or During Episode | Small mode peak area (a.u.) | Large mode peak area (a.u.) | Small mode diameter (nm) | Large mode diameter (nm) | Peak area ratio During : Before | Ratios of cube of mode diameter During : Before |
|---|---|---|---|---|---|---|---|---|
| **IR-1** | Org | Before | 312 | 2621 | 178 | 509 | 2.84 (small mode) | 2.10 (small mode) |
| | | During | 886 | 6415 | 228 | 603 | 2.45 (large mode) | 1.66 (large mode) |
| | SO4 | Before | 67 | 4051 | 183 | 589 | 6.44 (small mode) | 1.88 (small mode) |
| | | During | 431 | 11421 | 226 | 671 | 2.82 (large mode) | 1.47 (large mode) |
| **IR-3** | | Before | 255 | 2051 | 216 | 513 | 1.69 (small mode) | 1.09 (small mode) |
| | Org | During | 430 | 5808 | 223 | 531 | 2.83 (large mode) | 1.11 (large mode) |
| | | Before | 127 | 6358 | 222 | 570 | 1.54 (small mode) | 1.49 (small mode) |
| | SO4 | During | 195 | 10513 | 254 | 582 | 1.65 (large mode) | 1.06 (large mode) |
| **IR-5** | | Before | 165 | 2778 | 179 | 503 | 1.85 (small mode) | 1.54 (small mode) |
| | Org | During | 305 | 5983 | 207 | 555 | 2.15 (large mode) | 1.34 (large mode) |
| | | Before | 86 | 3880 | 198 | 585 | 1.97 (small mode) | 1.20 (small mode) |
| | SO4 | During | 170 | 9980 | 211 | 593 | 2.57 (large mode) | 1.04 (large mode) |

5. Page 10, Section 3.4 could be merged into Section 3.1 or just after the Section 3.1 (moving Section 3.2 to 3.3. . .).

Changed as suggested.

6. Page 12, Line 275: Please replace Figure 9c by Figure 9a.

Changed as suggested.

**Anonymous Referee #2**

**General comments**

The authors showed scientific findings on role of photochemistry in the evolution of secondary PM characteristics base on analyses of measurement data with high temporal resolution. The manuscript is generally well written and constructed, and therefore may be suitable for future publication in Atmospheric Chemistry and Physics after revisions based on referees' and other relevant comments. In addition, I recommend the authors to highlight the remarkable progress from their previous studies because they have already published several papers on the measurement data used in this study.

We have highlighted some key findings from our previous studies and the unique aspects of this study in the introduction.

Change from:

We conducted four one-month campaigns in each of the four seasons at the Hong Kong University of Science and Technology (HKUST) Air Quality Research Supersite (AQRS) from May 2011 to February 2012 using an Aerodyne HR-ToF-AMS for non-refractory PM1 (PM with aerodynamic diameter less than 1 micron). We have previously examined the general characteristics (Lee et al., 2013; Li et al., 2013, 2015), seasonal trends (Li et al., 2015), aqueous-phase process in the formation of oxygenated organics (Lee et al., 2013; Li et al., 2013, 2015), organic sulfur compounds (Huang et al., 2015), and particle hygroscopicity (Meng et al., 2014; Yeung et al., 2014). Most of these studies did not explore in detail the evolution of PM characteristics on the timescale of hours. The current work thus focuses on high PM episodes lasting a day or several days across the seasons. Specifically, we aim to investigate the role of local photochemical activity in the formation and evolution of secondary aerosols based on the real-time chemical composition and size distribution data.

To:

We conducted four one-month campaigns in each of the four seasons at the Hong Kong University of Science and Technology (HKUST) Air Quality Research Supersite (AQRS) from May 2011 to February 2012 using an Aerodyne HR-ToF-AMS for non-refractory $PM_1$ (PM with aerodynamic diameter less than 1 micron). In our previous studies, we found that photochemical oxidation during a haze episode

and aqueous-phase reactions during two foggy periods both led to a high degree of oxygenation of organics due to aging in gas phase and/or aqueous phase with substantial SOA formation (Lee et al., 2013 and Li et al., 2013). In spring and summer, SOA, with abundant SVOOA, was more likely to form locally. The oxygen-to-carbon atomic ratio (O:C) and average carbon oxidation state ($\overline{OS_c}$) peaked in the afternoon in spring and summer (Li et al., 2015). In autumn and winter, LVOOA dominated in SOA. The O:C ratio and $\overline{OS_c}$ showed little diurnal variation. Huang et al. (2015) estimated the contents of organic sulfur compounds in Hong Kong in September 2011. They highlighted the importance of both aqueous-phase processing and regional influence for the formation of organic sulfur compounds. Closure analysis was performed between the PM hygroscopicity measured by an hygroscopic tandem differential mobility analyzer (HTDMA) and chemical composition measured by an HR-ToF-AMS and a constant growth factor of 1.18 for organics was found to be adequate for a good closure, given the dominant contribution of the very hygroscopic sulfate at this suburban site (Cheung et al., 2016; Yeung et al., 2014). Meng et al. ( 2014) found that the aerosol hygroscopic parameter ($k$) decreased with an increasing organic-to-inorganic volume ratio. Furthermore, the concentration of cloud condensation nuclei (CCN) was found to be more sensitive to the mixing state and hygroscopicity of the particles at a high supersaturation (SS)=0.70% and a low SS=0.15%, respectively.

The above studies primarily focused on the analysis of campaign-average scenarios, without specifically looking at episodic events that occurred during the campaign.  In the current study, we investigated the chemical transformation and size evolution of secondary aerosols in high particulate matter episodes across the four seasons. Specifically, we examined in detail the photochemical evolution in a particular episode in which local influences were dominant. Results from the current study reveal the rapid evolution of secondary aerosols and are relevant to other megacities with large precursor input and high photochemical activity.

**Specific comments:**

1.  Page 6, Line 129-130: How can I find "no significant peaks of benzene and toluene after 10:00" from Fig. S3?

    We have added the time series of benzene and toluene in supplemental information as shown below.

[Figure]

2. Page 7, Line 147: Is the quite low fraction of POA even in cool season common in and around the study region, or specific at the study site (period)?

The HKUST supersite is located on the campus of HKUST, which sits on the hillside of Clear Water Bay on the east coast of Hong Kong that has relatively low population density. There is little local emission around the site. Two minor local sources include Clear Water Bay Road and a small student canteen. The Clear Water Bay Road is located outside of the campus and has mild traffic. The student canteen within 200m of the site, which operates only from 18:00LT (local time) until midnight. Therefore, it is not surprising that the POA fraction in our study is low even in cool season. As an indicator for primary PM, elemental carbon (EC) concentrations in $PM_{2.5}$ via off-line analysis after filter sampling at this site from March 2011 to February 2012 were found to be low throughout the year ($0.86 \pm 0.53 \mu g/m^3$) (Huang et al., 2014). Such description has been added in the revised manuscript.

3. Page 9. Line 197-199: If organics and sulfate are internally mixed, diameters of organics and sulfate are strongly correlated. However, is the strong correlation sufficient evidence for the internal mixing?

If the organics and sulfate are internally mixed (they exist in the same particle), their diameters should strongly correlate with each other and the growth rate of particle size should be similar (indicated by the slope of in Figure 5). The observed strong correlation and slope of unity (correlated in time and size) suggest that these species are likely internally mixed, although we cannot exclude out the possibility of external mixing. On the other hand, if the mode diameters of sulfate and organics did not change coherently with a strong correlation and a slope close to unity, these particles are more likely externally mixed. Using such correlations in size to indicate mixing states of species has been applied in previous field studies such as by Bahreini et al. (2003). It should be noted that these conditions of correlation and

slope are necessary but not sufficient evidence for internal mixing. In the original text, we stated that the large mode diameters of organics and sulfate (Pearson's R value equals to 0.7) were strongly correlated with a slope close to unity in LWC episodes (Figure 5), suggesting that organics and sulfate are likely internally mixed in large particles. We did not exclude the possibility of external mixing.

Pls also see our response to Comment #3 of Reviewer 1

Change from:

Various studies have analyzed the particle mixing state based on single-particle instruments such as the aerosol time-of-flight mass spectrometer (Healy et al., 2013, 2014; Yang et al., 2012) and the single-particle aerosol mass spectrometer (Wang et al., 2015). Particle mixing state can also be inferred from particle size information obtained with the AMS. The large mode diameters of organics and sulfate (Pearson's R value equals to 0.7) were strongly correlated with a slope close to unity in LWC episodes (Figure 5), suggesting that organics and sulfate are likely internally mixed in large particles.

To:

Various studies have analyzed the particle mixing state using single-particle instruments such as the aerosol time-of-flight mass spectrometer (Healy et al., 2013, 2014; Yang et al., 2012) and the single-particle aerosol mass spectrometer (Wang et al., 2015). Particle mixing state can also be inferred from particle size information obtained with the AMS. If the organics and sulfate are internally mixed (i.e. they exist in the same particle), their diameters should be strongly correlated with each other and their size should grow at a similar rate. The observed strong correlation and slope of unity (correlated in time and size) suggest that these species are likely internally mixed, although we cannot completely exclude the possibility of external mixing. On the other hand, if the mode diameters of sulfate and organics did not change coherently and exhibit a strong correlation with a slope close to unity, these particles were more likely externally mixed. Bahreini et al. (2003) have used such correlations in size to indicate the mixing states of species. In our study, the large mode diameters of organics and sulfate were strongly correlated (Pearson's R value equals 0.7) with a slope close to unity in LWC episodes (Figure 5), suggesting that organics and sulfate were likely internally mixed in the large particles. However, these conditions of correlation and slope are necessary but not sufficient evidence for internal mixing.

4. Page 9, Line 199: Does the authors consider that fraction of sulfate was higher in LWC episodes (56%)

than the other episodes because of efficient aqueous oxidation?

Although the fraction of sulfate was higher in LWC episodes than in other episodes (LWC: 56%, IR:49%, and LRT: 45%), the absolute mass concentrations of sulfate were quite similar (LWC: 12.4 µg m$^{-3}$, IR:11.7 µg m$^{-3}$ and LRT:11.8 µg m$^{-3}$). Aqueous oxidation is considered to be important in LWC episodes, when the aerosol liquid water content was relatively high and solar irradiance was quite low (Table 1). However, we did not aim to compare the efficiency of different pathways of sulfate formation under different conditions. We apologized that we may have confused the reviewer in the manuscript.

5. Page 10, Line 214-217: Please clarify how the percentages were estimated. In Fig. 5, the range of sulfate particle diameter seems to be smaller in IR episodes than LWC episodes.

The percentage numbers were calculated by comparing the smallest diameter in the blue shaded area and the largest diameter in the orange shaded area in each episode. Then, the percentage changes were averaged in each episode type. For example, there are two episodes in the LWC episode type, and the percentage changes for mode diameters in these two episodes were averaged to obtain the percentage changes for the LWC episode type.

The small range of sulfate particle diameter in Figure 5 is because only data points DURING the episodes were plotted. The size variation DURING the episodes were less obvious compared to days BEFORE the episodes (as illustrated in Line 222 of the manuscript). Please also refer to our response to Comment #4 of Reviewer 1.

Change from:

Figure 6 shows that, in the two LWC episodes, the size variations for both the small and large modes were less obvious: -2.5% for organics small mode, +8.1% for organics large mode, +1.6% for sulfate small mode, and -3% for sulfate large mode from start to end as shown in the figure. By contrast, the size variations in the IR episodes were much more drastic: +51.3% for organics small mode, +40.5% for organics large mode, +45.4% for sulfate small mode, and +35.9% for sulfate large mode.

To:

Figure 6 shows the particle mass mode diameters and areas (concentrations) in the LWC and IR types of episodes. We obtained the percentage changes in mode diameters by comparing the smallest diameter before the episode and the largest diameter during the episode for each episode. These percentages in

each episode was then averaged to obtain the percentage changes for each episode type. The results show that the changes in mode diameter were small in the LWC episodes: -2.5% for small mode organics, +8.1% for large mode organics, +1.6% for small mode sulfate, and -3% for large mode sulfate. In contrast, the changes in mode diameters changes were much more drastic in the IR episodes: +51.3% for small mode organics, +40.5% for large mode organics, +45.4% for small mode sulfate, and +35.9% for large mode sulfate. Furthermore, particle size usually increased more rapidly before the IR episodes (shaded in blue in Figure 6) than during the episodes (shaded in orange).

6. Page 10, Line 235: Can discussion on the "typical" IR episode (E4) be generally applicable to the other four IR episodes in different season? Although temporal variation patterns in E4 may be similar to those in Fig. 2b, temporal variation patterns and ratio of LVOOA: SVOOA seems to be quite different from one IR episode to another (Fig. S4).

It was a mistake that we describe E4 as "typical". It is corrected in the revised manuscript.

We chose E4 for a detailed study because of its land-sea breeze wind pattern and the low wind speeds, which provide a relatively stagnant condition with less influence of transported pollutants for a close examination of in-situ photochemical evolution. Although other IR episodes had high solar irradiance, they are complicated by regional transport effects. It is not appropriate to directly apply all the conclusions from E4 to the other IR episodes.

7. Page 10, Line 238-243: Does this mean that 72-h backward trajectory-based air mass origin could not explain the ground-level pollution, or the trajectories (at least for the first 24h) reflected a land-sea breeze pattern?

We used the overall trajectory analysis and further clustering to distinguish air masses into long-, medium-, and short-range transport patterns in groups. It is difficult to reflect a land-sea breeze pattern from these analyses. Rather, a detailed examination of the hourly backward trajectories during E4 (as below) shows that the air mass shifted from inland region to oceanic region, reflecting the land-sea breeze pattern. See figures below.

[Figure]

[Figure]

8.  Page 11, Line 274-277: SOR was the maximum after 18:00 on the day. Is SOR appropriate indicator of the photochemical SO2 conversion?

The sulfur conversion ratio has been widely used to examine the atmospheric formation of particulate sulfur oxidation products from $SO_2$ (Khoder, 2002; Lee et al., 2015; Miyakawa et al., 2007; Squizzato et al., 2013; Sun et al., 2006; Wang et al., 2005).  SOR was at maximum after 18:00 because of the low level of $SO_2$ gas. During daytime, especially from 10:00 to 18:00, particle-phase sulfate increased while $SO_2$ decreased. Moreover, the increases in SOR coincided with the increase in the ratio of benzene to toluene (Figure 9c), which is an indication of photochemical age. Sulfate/$\Delta$CO ratio also increased as photochemical age increased (Figure 10).  We consider SOR an appropriate indicator of photochemical conversion of $SO_2$ in this analysis.

[Figure]

Time of Day

9. Technical corrections: Page 6, Line 129: "benzenze" => "benzene"

Changed as suggested.

10. Page 10, Line 224-232: "3.4. Frequency of high PM1 episodes" can be moved into "3.1. Meteorological conditions and classification of episodes".

Changed as suggested.

11. Page 13, Line 313: "corresonding" => "corresponding"

Changed as suggested.

12. Fig. S4: Why pressure decreases with increasing mixing height? It is hard to believe that pressure was frequently less than 900 hPa at the ground-level of the study site. Air temperature in 05/14-05/16 episode seems to be quite low. Is it correct? Please provide time series of gaseous species other than OX (i.e., CO, SO2 and NOX).

The pressure we used was the average pressure in the boundary layer, not the surface pressure. We have removed the pressure in the revised manuscript. After comparison with other data, we indeed found that there is a deviation of the HKUST supersite temperature with the other temperature measurements within this region. We have changed the temperature data from HKUST supersite to the temperature data from an adjacent site (HKUST pier, 100 m away) as shown below. We have also added the time series of gaseous species (CO, $NO_x$, $SO_2$, and $O_x$) in the revised supplementary information.

[Figure]

[Figure]

**HKUST Supersite   HKUST Tower B   HKUST Pier   Other sites**

**Reference:**

Bahreini, R., Jimenez, J. L., Wang, J., Flagan, R. C., Seinfeld, J. H., Jayne, J. T. and Worsnop, D. R.: Aircraft-based aerosol size and composition measurements during ACE-Asia using an Aerodyne aerosol mass spectrometer, J. Geophys. Res., 108(D23), 8645, doi:10.1029/2002JD003226, 2003.

Cheung, H. H. Y., Tan, H., Xu, H., Li, F., Wu, C., Yu, J. Z. and Chan, C. K.: Measurements of non-volatile aerosols with a VTDMA and their correlations with carbonaceous aerosols in Guangzhou, China, Atmos. Chem. Phys., 16(13), 8431–8446, doi:10.5194/acp-16-8431-2016, 2016.

Huang, D. D., Li, Y. J., Lee, B. P. and Chan, C. K.: Analysis of Organic Sulfur Compounds in Atmospheric Aerosols at the HKUST Supersite in Hong Kong Using HR-ToF-AMS, Environ. Sci. Technol., 150305134049007, doi:10.1021/es5056269, 2015.

Huang, X. H. H., Bian, Q., Ng, W. M., Louie, P. K. K. and Yu, J. Z.: Characterization of PM2.5 major components and source investigation in suburban Hong Kong: A one year monitoring study, Aerosol Air Qual. Res., 14(1), 237–250, doi:10.4209/aaqr.2013.01.0020, 2014.

Khoder, M. I.: Atmospheric conversion of sulfur dioxide to particulate sulfate and nitrogen dioxide to particulate nitrate and gaseous nitric acid in an urban area, Chemosphere, 49(6), 675–684, doi:10.1016/S0045-6535(02)00391-0, 2002.

Lee, B. P., Li, Y. J., Yu, J. Z., Louie, P. K. K. and Chan, C. K.: Physical and chemical characterization of ambient aerosol by HR-ToF-AMS at a suburban site in Hong Kong during springtime 2011, J. Geophys. Res. Atmos., 118(15), 8625–8639, doi:10.1002/jgrd.50658, 2013.

Lee, B. P., Li, Y. J., Yu, J. Z., Louie, P. K. K. and Chan, C. K.: Journal of Geophysical Research : Atmospheres, , 1–19, doi:10.1002/2015JD023311.Received, 2015.

Li, Y. J., Lee, B. Y. L., Yu, J. Z., Ng, N. L. and Chan, C. K.: Evaluating the degree of oxygenation of organic aerosol during foggy and hazy days in Hong Kong using high-resolution time-of-flight aerosol mass spectrometry (HR-ToF-AMS), Atmos. Chem. Phys., 13(17), 8739–8753, doi:10.5194/acp-13-8739-2013, 2013.

Li, Y. J., Lee, B. P., Su, L., Fung, J. C. H. and Chan, C. K.: Seasonal characteristics of fine particulate

matter (PM) based on high-resolution time-of-flight aerosol mass spectrometric (HR-ToF-AMS) measurements at the HKUST Supersite in Hong Kong, Atmos. Chem. Phys., 15(1), 37–53, doi:10.5194/acp-15-37-2015, 2015.

Meng, J. W., Yeung, M. C., Li, Y. J., Lee, B. Y. L. and Chan, C. K.: Size-resolved cloud condensation nuclei (CCN) activity and closure analysis at the HKUST Supersite in Hong Kong, Atmos. Chem. Phys., 14(18), 10267–10282, doi:10.5194/acp-14-10267-2014, 2014.

Miyakawa, T., Takegawa, N. and Kondo, Y.: Removal of sulfur dioxide and formation of sulfate aerosol in Tokyo, J. Geophys. Res. Atmos., 112(13), 1–13, doi:10.1029/2006JD007896, 2007.

Squizzato, S., Masiol, M., Brunelli, a., Pistollato, S., Tarabotti, E., Rampazzo, G. and Pavoni, B.: Factors determining the formation of secondary inorganic aerosol: A case study in the Po Valley (Italy), Atmos. Chem. Phys., 13(4), 1927–1939, doi:10.5194/acp-13-1927-2013, 2013.

Sun, Y., Zhuang, G., Tang, A., Wang, Y. and An, Z.: Chemical Characteristics of PM 2.5 and PM 10 in Haze−Fog Episodes in Beijing, Environ. Sci. Technol., 40(10), 3148–3155, doi:10.1021/es051533g, 2006.

Wang, Y., Zhuang, G., Tang, A., Yuan, H., Sun, Y., Chen, S. and Zheng, A.: The ion chemistry and the source of PM2.5 aerosol in Beijing, Atmos. Environ., 39(21), 3771–3784, doi:10.1016/j.atmosenv.2005.03.013, 2005.

Yeung, M. C., Lee, B. P., Li, Y. J. and Chan, C. K.: Simultaneous HTDMA and HR-ToF-AMS measurements at the HKUST Supersite in Hong Kong in 2011, J. Geophys. Res. Atmos., 119(16), 9864–9883, doi:10.1002/2013JD021146, 2014.

---

## Author Response (AR2)

**Comment:**

1. Page 9, Line 205-206: it is interesting that in Figure 3, OA in LWC episodes fall into a lower O:C region than those in IR episodes. Does this indicate that the photo-oxidation is more effective to form higher oxidized organics than the aqueous oxidation? While as shown in Fig. 9 of Li et al. (2016), fog, cloud, and rain droplets often contain many highly oxygenated, high molar mass, and low-volatility compounds, indicating aqueous- phase processing can form highly oxidized compounds (Mazzoleni et al., 2010; Zhao et al., 2013). It is helpful to discuss the differences in oxidation state due to photo/aqueous oxidation and compare the oxidation state in this study with previous studies.

**Response:**

We thank the co-Editor for the suggestion to clarify this in the manuscript.

A number of studies (Li et al., 2016; Mazzoleni et al., 2010; Zhao et al., 2013) show that highly oxygenated organic compounds can be formed via aqueous-phase processing. Field measurements as in our case, however, are complicated by primary emissions for analysis of OA during foggy periods because foggy periods were generally associated with stagnant air and accumulated primary emissions. In Y.J. Li et al.(2013), the degree of oxygenation for the overall OA during photochemical oxidation was indeed higher than that in foggy periods characteristic of the aqueous-phase processing. Specifically, we found that the order of degree of oxygenation in this campaign: Hazy > Foggy > Non-fog/Non-haze from the AMS analysis.

However, this might not necessarily mean in general that aqueous processing produces less oxidized OA than photochemical oxidation does. The complication is that a lot of the oxidized OA during foggy periods was scavenged into fog droplets, while AMS only captures the sub-micrometer particles. Even after drying, the AMS will only sample interstitial particles and probably a very small portion of dried fog droplets. In other words, more oxidized species from aqueous-phase processing, if present, might not be captured by AMS during foggy periods. On the other hand, those SOA in hazy periods (mostly photochemical oxidation) are most likely in the sub-micrometer range and are effectively captured by AMS. With these complications, we hesitate to make general statements on how the aging process affects the degree of oxygenation of SOA.

[revised manuscript text omitted]